# Response of Runoff Change to Extreme Climate Evolution in a Typical Watershed of Karst Trough Valley, SW China

**Luhua Wu** [1,2,3,*], **Dan Chen** [1], **Dongni Yang** [1], **Guangjie Luo** [4], **Jinfeng Wang** [5] **and Fei Chen** [6]

1    School of Economics and Management, Tongren University, Tongren 554300, China;
     chen2514922405@126.com (D.C.); dong2632912301@126.com (D.Y.)
2    Tongren Rural Revitalization Research Institute, Tongren 554300, China
3    Fanjing Mountain National Park Research Institute, Tongren 554300, China
4    Guizhou Provincial Key Laboratory of Geographic State Monitoring, Guiyang 550018, China;
     luoguangjie@gznc.edu.cn
5    School of Economics and Management, Liupanshui Normal University, Liupanshui 553004, China;
     jfwanggz@126.com
6    Guizhou Institute of Water Resources Science, Guiyang 550002, China; chenfeicn@126.com
*    Correspondence: jgywlh@gztrc.edu.cn

**Abstract:** Identifying the response of runoff changes to extreme climate evolution was of great scientific significance for the rational regulation of watershed water resources and the prevention of hydrological disasters. However, the time–frequency response relationships were not clear. The Yinjiang River watershed, a typical watershed with karst trough valley areas, was chosen to identify the impact of different climatic driving factors on runoff changes from 1984 to 2015. Continuous wavelet transform (CWT), cross-wavelet transform (XWT), and wavelet coherence transform (WTC) were performed to study the response relationship and time–frequency effect between runoff changes and extreme climate change at different time scales. The main results showed that: (1) Twelve extreme climate indices (ECIs) were detected to have a significant impact on runoff changes, mainly on a 6-year time scale; (2) The R10 and Rx1day in extreme precipitation index and SU34.4 and TNx in the extreme temperature index were the main driving factors of runoff changes, which had relatively large impacts on runoff changes in high and low energy vibration regions. However, the remaining eight ECIs that passed the 0.05 confidence level showed relatively large impacts on runoff changes only in low energy vibration regions; (3) The transition of the interaction between ECIs and runoff changes in high and low time–frequency scales was related to the abrupt change characteristics of the ECIs. The correlation of abrupt change was an important reason for the emergence of highly correlated regions that trigger high and low energy vibrations; (4) As a whole, the extreme precipitation events were ahead of runoff changes at the high time–frequency scale and exhibited small lag effects at the low time–frequency scale, while extreme temperature events were mainly ahead of runoff changes. This study has effectively revealed the impact of climate factors at different scales on runoff changes, and provides a theoretical understanding for regulating and managing water resources in karst basins.

**Keywords:** karst; watershed; runoff change; extreme climate evolution; wavelet analysis

## 1. Introduction

In the context of global warming, extreme climate change has become a major problem that restricts the development of human society [1]. Under the combined action of various factors, extreme climate events have aroused widespread concern [2], especially on matters concerning the intensification of the water cycle, the intensification of extreme climate events, and the increase of drought and flood frequency. These events have seriously affected the rational regulation of watershed water resources and the prevention of hydrological disasters. In recent years, the frequency of extreme precipitation has increased along with the rising number of floods, droughts, and other disasters caused by

extreme precipitation [3,4]. The frequency and amplitude of extreme temperature events have generally increased, and they have compounded the responses of eco-hydrological processes [5–11]. Runoff mainly manifests as the combined effects of climate change and underlying surface conditions [12], and the most direct impact of climate change on runoff is the change in runoff size and spatial and temporal distribution [13]. The generation of a runoff is closely related to changes of PT, temperature, and evaporation. According to the IPCC (Intergovernmental Panel on Climate Change) reports, the increase in temperature has caused the spatial and temporal redistribution of regional water resources and the evolution of extreme climate events [14], which greatly affects ecological hydrological processes and regional water resources, and further seriously threatens the regulation and safety management of water resources in karst basins. Therefore, a reliable assessment of the response of runoff to extreme climate events is crucial in developing strategic plans for the sustainable safety management of water resources in watersheds.

The spatiotemporal evolution process and driving mechanism of dry hot composite extreme events and quantitative risk assessment of extreme climate change on temperature and runoff have received significant attention in recent studies [15–18]. The existing research on runoff and extreme climate is usually conducted at the administrative regional level, but there is a lack of attention to watershed scale, especially in karst basins affected by geological structure and geomorphic evolution where the time–frequency response relationships are not clear [19–21]. In addition, most studies have focused on analyzing the correlations among the influencing factors of extreme climate events or runoff evolution by using linear trends and multiple linear or nonlinear regression [22–24], but these methods cannot show the correlation and phase difference between runoff and extreme climate evolution in the time domain. From the viewpoint of research content, these studies have emphasised the following topics: the characteristics, causes, and mechanisms of the evolution of the extreme temperature index (ETI) and the extreme precipitation index (ERI) on multi-spatial scales [25,26], and the uncertainty of extreme climate evolution and variability of time–space differences, occurrence frequency, amplitude, and instability, among others [27–29]. Hydrological processes are extremely sensitive to climate change, especially the response to extreme climate events [30]. However, little research has been conducted on the multi-temporal correlation between runoff and extreme climate events. Hydrologists have recently focused on the inter-annual and multi-decadal periodicities of the largest rivers in the world through mouth-to-ocean discharge estimations, and their relationship with global climate indices was explored along with the relationship of multi-temporal scales of runoff changes and meteorological elements of basins [31,32]. Karst watersheds are characterized by thin surface soil, high infiltration capacity, and complex topography due to the special geologic condition and thus generally are distinct from non-karst watersheds [33–35]. However, there is a lack of attention to the multi-scale coupling study of extreme climate events and ecological hydrological processes at the watershed scale. These challenges prevent government decision-makers from effectively regulating the water resource security risks caused by extreme climate events.

Wavelet analysis has become a powerful technique in studying geophysical processes or signals [26–37]. This method can help researchers localise power variations within a time series by decomposing data into time-scale space. Wavelet analysis is ideal for analysing non-stationary signals and identifying short- and large-scale periodic phenomena. In the field of hydrology, continuous wavelet transform (CWT) has been recently used to determine the effect of climatic phenomena on stream flow regimes [10,11,38] or runoff processes [39]. CWT has also been widely used for studying the hydrogeological behaviour of karst systems [35]. Hydrogeological responses in the format of physicochemical time series have also been explored to investigate transport properties, turbidity dynamics [40], and temperature–runoff relationships during snowmel and groundwater variations in geological contexts [41]. Surface–groundwater interactions have been studied by using CWT to improve the understanding of river flow components in karst environments [42]. Meanwhile, cross-wavelet transform (XWT), which has strong signal coupling and resolution

ability, can show the common high energy time–frequency region and phase correlation of two time-series data. However, XWT has a great unsolved shortcoming that cannot find significant coherence when analysing the low energy time–frequency regions of two time series data in the time–frequency domain, and its functional defects in the low energy area must be compensated by wavelet coherence transform (WTC) [43]. The implications of climate and anthropic pressures on the short- to long-term changes in the water resources of a Mediterranean karst were assessed by using wavelet analysis [44]. The non-stationary relationships of ocean and atmosphere mean conditions and freshwater discharge, which were integrated at the continental scale, were studied by using cross-wavelet analysis [30]. The impacts of precipitation, air temperature, and evapotranspiration on an annual runoff in the source region of the Yangtze River were investigated in the time domain by using wavelet analysis and multiple regression [45]. Wavelet coherence analysis was used to determine the overall and scale-dependent similarities of temporal patterns of soil moisture in the karst catchments of south-western China [46]. CWT analysis was used to detect the trends and periodicity in sediment discharge, whilst wavelet coherence analysis was used to detect the temporal covariance between sediment discharge and water discharge, precipitation, potential evapotranspiration, and vegetation index in two typical karst watersheds in southwest China [35]. However, the knowledge is generally lacking in terms of the multiple time-scale effects of extreme climate events on runoff evolution in typical basins of karst trough valleys.

The Yinjiang River watershed, which was a typical watershed with karst trough valley areas located in southwest China, was chosen to firstly investigate the influence of extreme climate events on runoff evolution at different time scales. Then, hydrometeorological variables were used as input and output signals to diagnose the responses of runoff changes to the extreme climate change of karst watershed, in which the input variables were extreme climate index related to runoff change and the output variables were runoffs. Pearson correlation, CWT, XWT, and WTC were performed to reveal the response relationship between extreme climate change and runoff evolution in the Yinjiang River watershed based on time scales. The study objectives were as follows: (1) to diagnose extreme climatic events with significant impacts on runoff changes in a typical watershed of karst trough valley areas; (2) to identify the response characteristics of runoff changes to extreme temperature events and extreme precipitation events, respectively; and (3) to clarify the coupling relationships, phase relations, and temporal variability between significantly extreme climatic events and runoffs in common high and low energy time–frequency regions at different time scales. This study can provide theoretical support for the optimal regulation and safety management of water resources in karst basins and Guizhou Province against the background of extreme climate change.

## 2. Study Site

The Yinjiang River watershed (108°21′21″–108°47′27″ E, 27°53′17″–28°13′57″ N) located in the northeast of Guizhou Province (Figure 1a) was a typical karst valley watershed, SW China. The watershed was an important branch of the Wujiang River with an area of 691.56 km$^2$ in which the karst area was 376.77 km$^2$ and the non-karst area was 314.79 km$^2$, accounting for 54.68% and 45.32% of the total watershed area, respectively. The watershed decreased from southeast to northwest, ranging in a large scope with an elevation range of 439–2466 m above sea level and a mean elevation of 1033 m (Figure 1b). The watershed was located in the humid subtropical monsoon climate zone with an annual mean temperature of 17.14 °C (1984–2015); the maximum daily temperature was 40.4 °C, and the minimum daily temperature of −4.1 °C. The annual average precipitation was 1103.44 mm/year with a maximum precipitation 151.5 mm/day and an annual average evaporation of 667.01 mm/year based on the 32-year period (1984–2015).

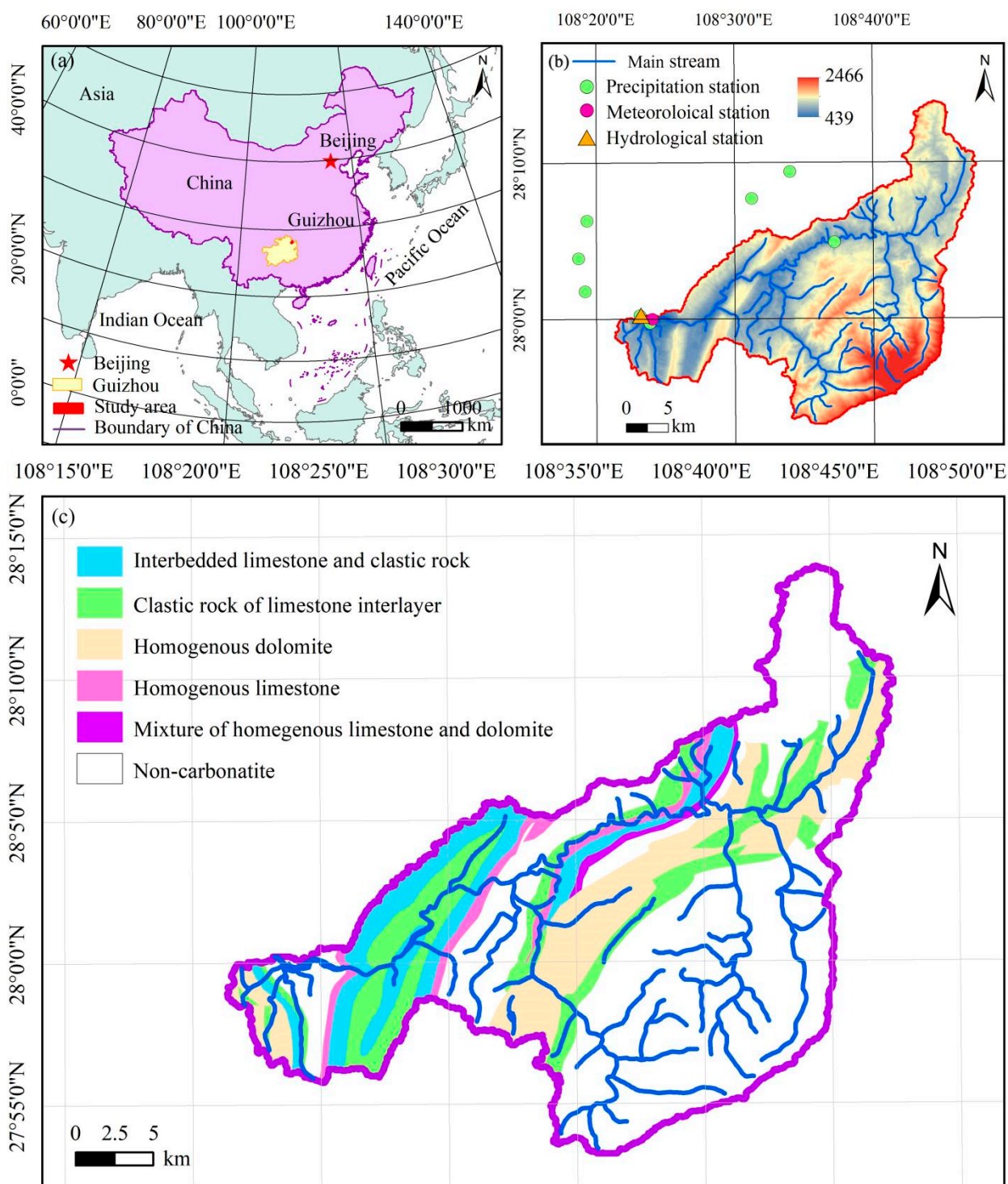

**Figure 1.** The location of study area in China (**a**), DEM (**b**), and lithology (**c**) in the Yinjiang River watershed.

The centre of the watershed was a long strip shaped valley formed by flowing water, with a syncline structure in the centre and steep slopes on both sides, which was mainly controlled by karst structures and lithology. The drainage basin was a typical combination of trough shaped landforms, with a narrow trough bottom, wide and high mountains on both wings, and developed micro karst landforms such as stone gullies, stone buds, and peak clusters on the slope surface. The basin was jointly constrained by eight geological backgrounds and its surface topography was controlled by six lithological coupling factors.

The karst area in the Yinjiang River basin was mainly controlled by homogenous dolomite, clastic rock of limestone interlayer, and interbedded limestone and clastic rock (Figure 1c), with controlled areas of 108.29 $km^2$, 106.81 $km^2$, and 69.12 $km^2$, respectively, accounting for 15.66%, 15.44%, and 9.99% of the total area (Table 1). Therefore, its hydrological process exhibited unique and significant differences compared to other karst basins. Its steep and fragmented surface, numerous underground spaces with cracks, and undulating terrain resulted in a small effective catchment area, low runoff coefficient, and serious losses of underground runoff and rainfall. The mean annual runoff was only $4.62 \times 10^8$ $m^3$/year based on the 32-year period (1984–2015). The Yinjiang River, which was the only river and the main channel that existed in the valley area and originated in Fanjing Mountain with abundant water resources, flowed from the eastern part of the basin to the bottom of the trough valley and eventually flowed out from the southwest of the watershed. Many residential areas and farmlands with the largest intensity of human activities were located on both sides of the river.

**Table 1.** Geological and lithological characteristics of the Yinjiang River watershed.

| Geological Stratums | Lithology | Area ($km^2$) | Proportion (%) |
|---|---|---|---|
| Combined layer of Liangshan, Qixia, and Maokou formation | Homogenous limestone | 23.79 | 3.44 |
| Heshan formation | Interbedded limestone and clastic rock | 69.12 | 9.99 |
| Jialing River formation | Clastic rock of limestone interlayer | 106.81 | 15.44 |
| Combined layer from Majuchong to Xiushan formation | Non-carbonatite | 317.72 | 45.94 |
| Loushanguan formation | Homogenous dolomite | 108.29 | 15.66 |
| Maotian formation | Mixture of homegenous limestone and dolomite | 6.78 | 0.98 |

The extreme precipitation events occurred in summer and were accompanied by extreme high-temperature events. July was the most serious month with a maximum runoff of 155 $m^3$/month and a maximum monthly rainfall of 588.1 mm/month. In addition, the maximum monthly evaporation in August was as high as 164.1 mm/month. The river network was mainly developed in karst area. Due to the lithological characteristics, many tributaries developed into dry valley rivers, exhibiting drought in the autumn and winter dry seasons and severe flooding in the spring and summer rainy seasons. It was extremely easy to erode the bedding slope under the scouring of heavy rainfall or torrential rain and this led to river blockage and deposition. The river formed at the bottom of the trough valley was the only channel for drainage and there was no reservoir dam in the main channel of the watershed. It was easy to form secondary runoff with large discharge because of the special karst background, which posed a great threat to soil erosion on slopes and farmland on both sides of the channel at the bottom of the trough valley.

## 3. Materials

In this study, the monthly runoff data of a hydrological station and daily precipitation data with the same length of eight rainfall observation stations for the study area were collected from the Guizhou Provincial Hydrology and Water Resources Bureau (http://www.gzswj.gov.cn/hydrology_gz_new/index.phtml) (accessed on 15 September 2016). Some of the average daily precipitation data for the watershed were interpolated with the Kriging method using ArcGIS 10 (ESRI 2010) with daily precipitation data of the rainfall observation stations in the watershed together with those from the neighbouring observation stations outside it. The daily data of maximum temperature, minimum

temperature, average temperature, and rainfall in the Yinjiang River watershed from 1984 to 2015 were selected from the China Meteorological Data Sharing Service System (http://cdc.cma.gov.cn/) (accessed on 1 March 2018). The missing data were less than 0.01%. The missing data were filled by the mean values of the data of the same station in the same period. This treatment of missing data did not affect the long-term trend of extreme climate [47]. Among them, the temperature data were homogenized and corrected by SNHT [48] and TPR methods [49]. Data quality control was accomplished by using the R software package "RClimDex" (http://etccdi.pacificclimate.org/software.shtml). All meteorological data had high accuracy, long observation duration, and complete data value. In addition, it was difficult to judge the breakpoint of heterogeneity in the homogeneity test of rainfall data because of the large noise of daily rainfall. Therefore, we first tested the homogeneity of monthly data after logarithmic transformation, and then judged the homogeneity of daily rainfall data [50]. The test of uniformity was accomplished by using the package "RHtestsV4" (http://etccdi.pacificclimate.org/software.shtml) (accessed on 13 April 2018). The result showed that there is no homogeneous breakpoint in all the data. To facilitate calculation and comparison, all the time series of extreme climate indices (ECIs) were standardized before analysis.

## 4. Methodology

### 4.1. Selection and Threshold Determination of ECI

The Expert Team for Climate Change Detection Monitoring and Indices (ETCCDMI) had defined 11 extreme precipitation indices (EPIs) and 16 extreme temperature indices (ETIs). On the basis of the climate characteristics of the Yinjiang River watershed, twenty-six ECIs defined by ETCCDMI and five ECIs defined by threshold were calculated. Therefore, the ECIs calculated in this study included 11 rainfall indices and 20 temperature indices. The definitions of these indices are shown in Tables 2 and 3, respectively.

This study assumed that a temperature variable had $n$ values. Then, the $n$ records were arranged in ascending order as $x_1, x_2, \ldots, x_m, \ldots x_n$. The percentile value was

$$x_0 = (1 - a)x_j + ax_{j+1} \tag{1}$$

where $j = [p(n + 1)]$ and $\alpha = p(n + 1) - j$. $j$ was the ordinal number of the air temperature records in ascending order, $p$ was the probability corresponding to the percentile value, and $n$ was a sequential sample size.

**Table 2.** Definitions of ETIs used in this study.

| ETIs | | | |
|---|---|---|---|
| **Identification** | **Indicator Name** | **Definitions** | **Unit** |
| FD0 | Frost days | Annual count when TN(daily minimum) < 0 °C | Day |
| FD0.6 | Number of frost days below 0.6 °C | Annual count when TN(daily minimum) < 0.6 °C, 0.6 °C is a user-defined threshold | Day |
| SU25 | Summer days | Annual count when TX(daily maximum) > 25 °C | Day |
| SU34.4 | Number of summer days above 34.4 °C | Annual count when TX(daily maximum) > 34.4 °C, 34.4 °C is a user-defined threshold | Day |
| ID0 | Ice days | Annual count when TX(daily maximum) < 0 °C | Day |
| ID6.4 | Number of ice days below 6.4 °C | Annual count when TX(daily maximum) < 6.4 °C, 6.4 °C is a user-defined threshold | Day |
| TR20 | Tropical nights | Annual count when TN (daily minimum) > 20 °C | Day |
| GSL | Growing season Length | Annual (1 January to 31 December in NH, 1 July to 30 June in SH) count between first span of at least 6 days with TG > 5 °C and first span after 1 July (1 January in SH) of 6 days with TG < 5 °C | Day |
| TR23.8 | Number of tropical nights above 23.8 °C | Annual count when TN(daily minimum) > 23.8 °C, 23.8 °C is a user-defined threshold | Day |

**Table 2.** *Cont.*

| ETIs | | | |
|---|---|---|---|
| **Identification** | **Indicator Name** | **Definitions** | **Unit** |
| DTR | Diurnal temperature range | Monthly mean difference between TX and TN | °C |
| TXx | Max Tmax | Monthly maximum value of daily maximum temp | °C |
| TNx | Max Tmin | Monthly maximum value of daily minimum temp | °C |
| TXn | Min Tmax | Monthly minimum value of daily maximum temp | °C |
| TNn | Min Tmin | Monthly minimum value of daily minimum temp | °C |
| WSDI | Warm spell duration indicator | Annual count of days with at least 6 consecutive days when TX > 90th percentile | Day |
| CSDI | Cold spell duration indicator | Annual count of days with at least 6 consecutive days when TN < 10th percentile | Day |
| TN10p | Cool nights | Percentage of days when TN < 10th percentile | Day |
| TX10p | Cool days | Percentage of days when TX < 10th percentile | Day |
| TN90p | Warm nights | Percentage of days when TN > 90th percentile | Day |
| TX90p | Warm days | Percentage of days when TX > 90th percentile | Day |

**Table 3.** Definitions of EPIs used in this study.

| EPIs | | | |
|---|---|---|---|
| **Identification** | **Indicator Name** | **Definitions** | **Unit** |
| CDD | Consecutive dry days | Maximum number of consecutive days with RR < 1 mm | Day |
| CWD | Consecutive wet days | Maximum number of consecutive days with RR ≥ 1 mm | Day |
| R10 | Number of heavy precipitation days | Annual count of days when PRCP ≥ 10 mm | Day |
| R17.3 | Number of very heavy precipitation days above 17.3 mm | Annual count of days when PRCP ≥ 17.3 mm, 17.3 mm is a user-defined threshold | Day |
| R20 | Number of very heavy precipitation days | Annual count of days when PRCP ≥ 20 mm | Day |
| SDII | Simple daily intensity index | Annual total precipitation divided by the number of wet days (defined as PRCP ≥ 1.0 mm) in the year | mm/day |
| R95p | Very wet days | Annual total PRCP when RR > 95th percentile | mm |
| R99p | Extremely wet days | Annual total PRCP when RR > 99th percentile | mm |
| PRCPTOT | Annual total wet-day precipitation | Annual total PRCP in wet days (RR ≥ 1 mm) | mm |
| Rx1day | Max 1-day precipitation amount | Monthly maximum 1-day precipitation | mm |
| Rx5day | Max 5-day precipitation amount | Monthly maximum consecutive 5-day precipitation | mm |

The "RClimDex" package was used to calculate the ECIs under the R platform [51], and the Mann–Kendall method was applied to calculate the change trend of them. Through detection, the percentile values of extreme temperature events in this study were 90% (10%) and 95% (5%), respectively. The upper threshold of daily maximum temperature was 34.4 °C and the lower threshold of daily maximum temperature was 6.4 °C. The upper threshold of daily minimum temperature was 23.8 °C and the lower threshold of daily minimum temperature was 0.6 °C. This study defined the minimum threshold that met the highest rainfall as 17.3 mm.

*4.2. Wavelet Analysis*

In this study, the Pearson correlation coefficient method was firstly used to identify extreme climatic events which had significant relationship with runoff variation; CWT was used to reveal the energy relationship between ECI and runoff evolution over multiple time scales [31,52]. XWT and WTC were used to reveal the energy distribution differences and phase relations between ECIs and runoff over a common high energy and low energy vibration region [37,53]. Although wavelet analysis was a kind of statistical relationship, it could reveal the internal relationship between various variables more deeply. So, it was an effective method with which to study the relationship between meteorological and hydrological factors with a strong physical mechanism. In this study, wavelet analyses (CWT, XWT, and WTC) were carried out using a free Matlab software package (Math works, Nat-ick, MA, USA) kindly provided by Grinsted et al. [31] (accessed on 12 September 2021) at http://noc.ac.uk/using-science/crosswavelet-wavelet-coherence. The package includes code originally written by C. Torrence and G. Compo, available at: http://paos.colorado.edu/research/wavelets/ (accessed on 12 March 2018), and by Breitenberger E. of the University of Alaska. In order to avoid the boundary effect and high frequency false information of the wavelet, the region within the wavelet-influenced vertebra body was the effective spectral value. The region surrounded by the black thick solid line represented the energy vibration that passed the test of the red noise standard spectrum at the 0.05 significance level. Phase change reflected the difference of response time of runoff to extreme climatic factors, indicating that regional climate has a certain sustainability on runoff.

## 5. Result Analysis

*5.1. Correlation between ECI and Runoff Change*

The influence of extreme climate on runoff in river basin systems could be evaluated according to the correlations between ECI and runoff. On the basis of the number of study samples for the Pearson correlation, 0.35 and 0.45 were determined as the critical coefficient values at the confidence levels of 0.05 and 0.01, respectively. For the ETIs (Table 4), only SU34.4, TNx, and TX90p passed the 0.05 confidence level. However, for the ERIs, R99p and Rx1day passed the 0.05 confidence level, whereas PRCPTOT, R10, R17.3, R20, R95p, Rx5day, and SDII passed the 0.01 confidence level. Among them, relatively high correlation coefficients were detected for PRCPTOT and R10 with runoff changes. These results indicated that extreme temperature events had a low impact on runoff, whereas extreme precipitation events could greatly affect runoff changes.

**Table 4.** The correlation characteristics of ECIs and runoff.

| ECIs | R | Sig | ECIs | R | Sig |
|------|------|-----|------|------|-----|
| CDD | 0.01 | N | ID6.4 | 0.03 | N |
| CWD | −0.14 | N | SDII | 0.57 | 0.01 |
| PRCPTOT | 0.68 | 0.01 | SU25 | −0.24 | N |
| R10 | 0.67 | 0.01 | SU34.4 | −0.38 | 0.05 |
| R17.3 | 0.49 | 0.01 | TN10p | 0.03 | N |
| R20 | 0.53 | 0.01 | TN90p | −0.26 | N |
| R95p | 0.55 | 0.01 | TNn | −0.13 | N |
| R99p | 0.4 | 0.05 | TNx | −0.37 | 0.05 |
| Rx1day | 0.38 | 0.05 | TR20 | −0.07 | N |
| Rx5day | 0.51 | 0.01 | TR23.8 | −0.32 | N |
| CSDI | −0.02 | N | TX10p | 0.13 | N |
| DTR | −0.09 | N | TX90p | −0.35 | 0.05 |
| FD0.6 | 0.13 | N | TXn | −0.12 | N |
| FD0 | 0.1 | N | TXx | −0.28 | N |
| GSL | 0.14 | N | WSDI | −0.27 | N |
| ID0 | 0.22 | N | | | |

Note: 'N' represents that a significant correlation between ECI and runoff does not pass the significance test at the 0.5 or 0.01 confidence levels. 'R' represents the correlation coefficient. 'Sig' represents the level of significance.

### 5.2. Responses of Runoff to ECI Changes

Many rivers in karst areas were rain-fed rivers, and the change in runoff was closely related to climatic factors. Extreme climate events, especially extreme precipitation events, had considerable effects on the hydrological processes of basins. To study the time–frequency response of runoff towards the extreme climate factors, the ECIs that passed the 0.05 confidence level were selected on the basis of the Pearson correlation analyses, including three ETIs (SU34.4, TNx, and TX90p) and nine EPIs (SDII, PRCPTOT, R10, R17.3, R20, R95p, R99p, Rx1day, and Rx5day). Time–frequency and phase correlation analyses were conducted to identify the areas in which the correlated climate factors and runoff (denoted by Q) played strong roles in commonly high and low energy time–frequency regions.

5.2.1. Time Frequency Characteristics of Evolution Process of Runoff and ECIs

As shown in Figure 2, the changes in energy distribution since 1994 that appeared with high energy periods at 4- to 8-year scales peaked in 2006, after which the energy distribution weakened. The low energy time–frequency region mainly appeared in the high time frequency at a 3-year time scale before 2004. Energy distribution was increased to a high time frequency after 2004, and shifted from a low frequency to a high time frequency after 2006 and finally decreased.

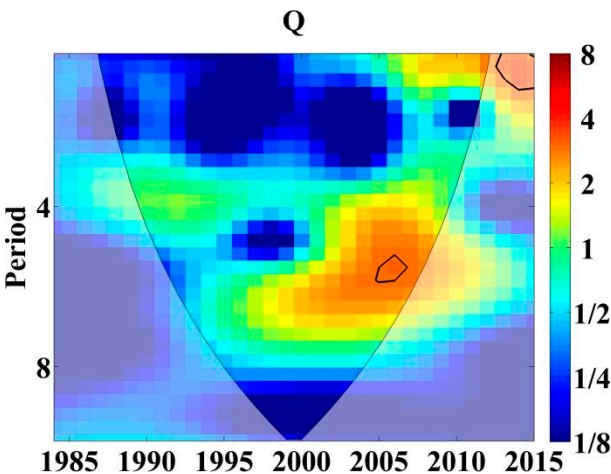

**Figure 2.** Continuous wavelet power spectra of runoff (Q) changes. The thick black contour designates the 5% significance level against red noise and the cone of influence (COI) where edge effects might distort the picture was shown as a lighter shade.

As shown in Figure 3, PRCPTOT, R99p, Rx1day, Rx5day, and TNx did not pass the red-noise standard spectral test at the 0.05 significance level, whereas the other ECIs passed the same test for some periods. For the ERI, high energy time–frequency regions mainly appeared at 1- to 2-year time scales, in which the years concentrated around 1995 and 1998, and then slightly weakened after 2000. For the ETI, the regions greatly varied over the entire period. Energy distribution was enhanced after 2000, and significant high energy areas mainly appeared around 2010. Over time, the energy distribution of Rx1day was shifted from high to low frequency in a decreasing trend until reaching the lowest time and frequency at a 6-year time scale around 2007. The energy distribution of R95p from 1993 to 1998 was concentrated in the high frequency region for the years depicted prior to the 5-year time scale. The energy distributions of SU34.4, TNx, and TX90p changed from low time frequency to high time frequency over time, whereas the energy distributions of the other EPIs (except Rx1day) tended to change from a high time frequency to a low time frequency in varying degrees. The tested results were not significant for the ECI in the entire period of continuously high and low time frequency regions, which indicated that the ECI varied greatly.

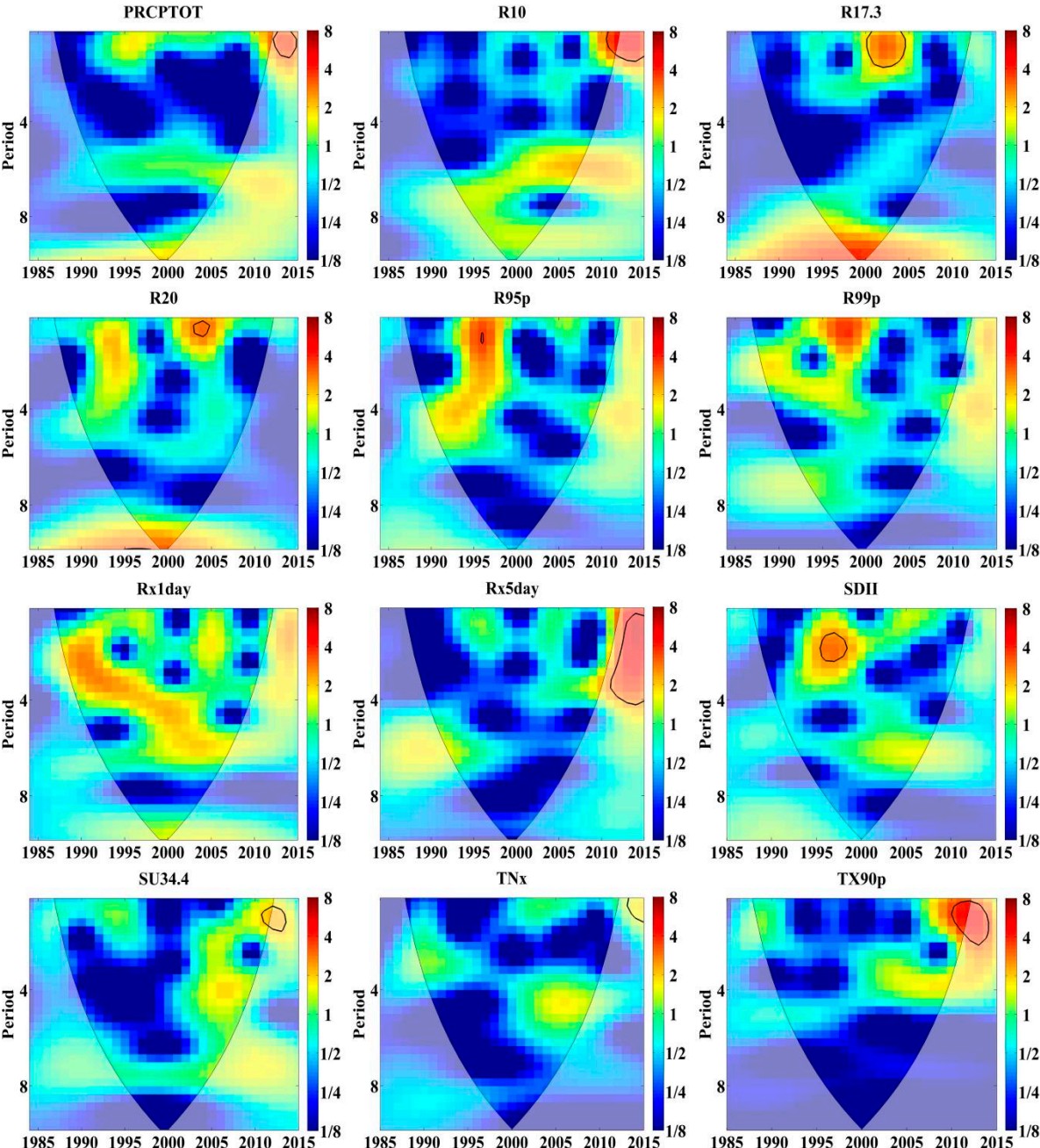

**Figure 3.** Continuous wavelet power spectra of the ECIs changes. The thick black contour designates the 5% significance level against red noise and the cone of influence (COI) where edge effects might distort the picture was shown as a lighter shade.

### 5.2.2. Phase Relation between Runoff and ETI

Figure 4 shows that SU34.4 was ahead of runoff change by 1–2 years at all time scales. SU34.4 and Q appeared in the high energy oscillation region at 4- to 5-year time scales from 2003 to 2006, and the phase angle was 45°, indicating that the variation of SU34.4 was ahead of the variation of runoff by approximately 1.5 years. After 2005, high energy oscillation regions appeared at 1- to 2-year time scales. The initial phase angle had increased gradually from 0° to 45°, indicating that the time–frequency relationship of SU34.4–Q had changed from being consistent to one that was ahead by approximately 1.5 years. The comparative results of WTC with XWT indicated a consistent correlation between high energy and low energy vibration regions at 4- to 5-year time scales (Figure 5). The obvious correlations

between common low energy time frequency regions were detected and had passed the red-noise standard spectral test at the 0.05 significance level.

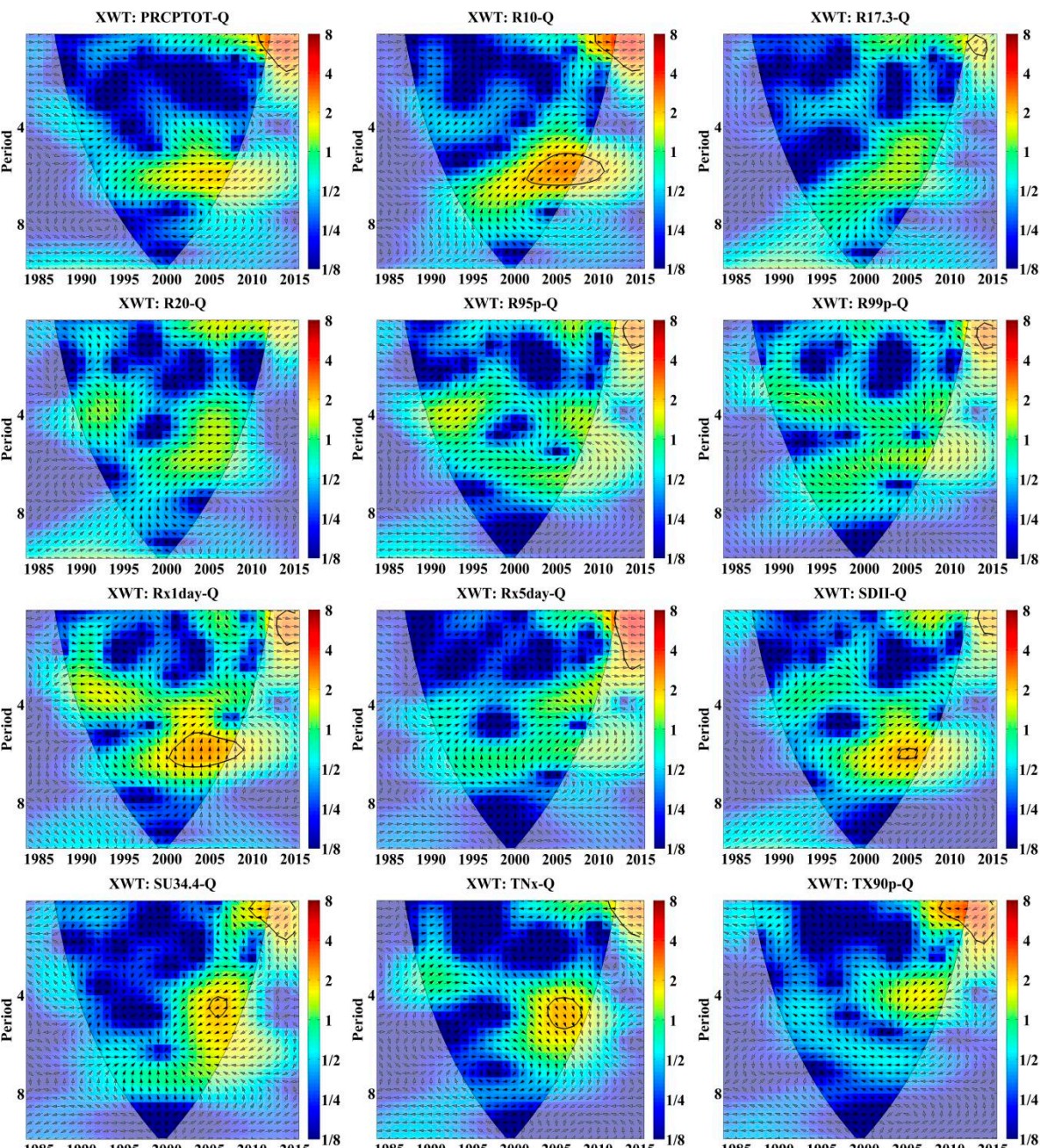

**Figure 4.** The wavelet power spectrum for ECIs and runoff (Q) in the Yinjiang River watershed from 1984 to 2015. The thick black contour designated the 5% significance level for red noise, and the cone of influence (COI) where edge effects might distort the picture was shown as a lighter shade. Phase change reflected the difference in response time of primary variables to influence factors. The phase relationship between influence factors and primary variables was indicated by arrows. The arrows from left to right indicated that the influencing factors and primary variables were in the same phase, which implied a positive correlation; the arrows from right to left indicated an inverse phase, which implied a negative correlation; the downward arrows indicated that influencing factors were 90° ahead of primary variables, and the upward arrows indicated that influence factors were 90° behind primary variables.

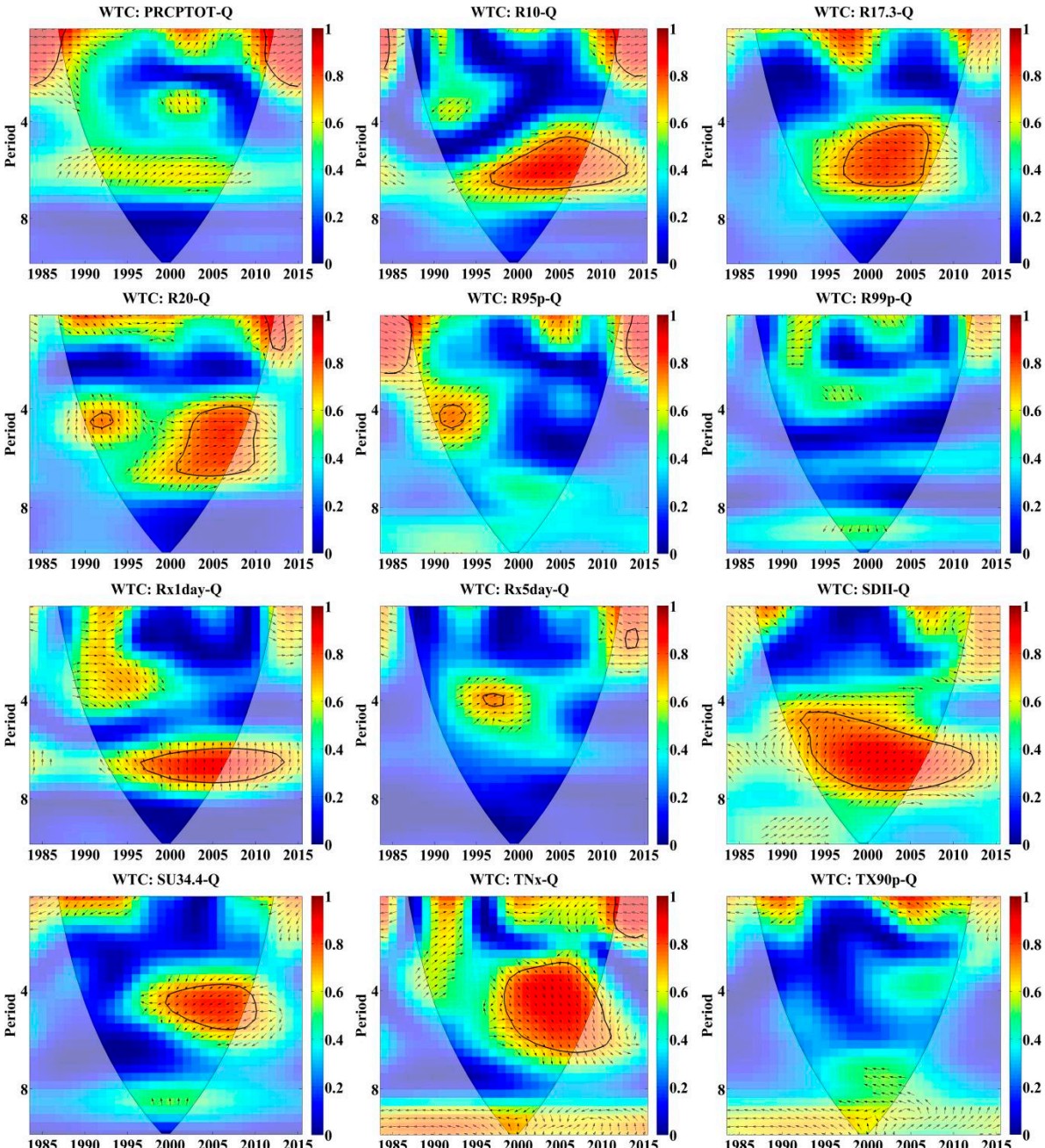

**Figure 5.** The wavelet condensation spectrum for ECIs and runoff (Q) in the Yinjiang River watershed from 1984 to 2015. The thick black contour designated the 5% significance level for red noise, and the cone of influence (COI) where edge effects might distort the picture was shown as a lighter shade. Phase change reflected the difference in response time of primary variables to influence factors. The phase relationship between influence factors and primary variables was indicated by arrows. The arrows from left to right indicated that the influencing factors and primary variables were in the same phase, which implied a positive correlation; the arrows from right to left indicated an inverse phase, which implied a negative correlation; the downward arrows indicated that influencing factors were 90° ahead of primary variables, and the upward arrows indicated that influence factors were 90° behind primary variables.

The time–frequency relationship with different phase correlation characteristics in high and low energy time frequency regions of TNx-Q and SU34.4-Q were consistent in terms of year and time scales. The phase correlation of TNx–Q was positive, and the phase

angle was approximately 60°, indicating that TNx was ahead of runoff by nearly 2 years in both high and low energy time frequency regions. In addition, the vibration relationship was relatively consistent at 1- to 2-year time scales after 2000, especially in the low energy time–frequency region. Moreover, the correlation was prominent, and the time scale range had expanded, which indicated a negative correlation. The phase angle increased from 0° in 2000 to 45° in 2007 and then gradually decreased to 0° in 2007.

The high energy time–frequency region of TX90p–Q was mainly concentrated in the high time–frequency scales at 1–2 year time scales, whereas the low energy time–frequency region was apparent in some high time–frequency and low time–frequency scales. It had found that no significant high energy vibration relation of TX90p–Q was detected at a 4-year time scale, but the low energy vibration relation was extremely significant. The phase angle of 60° suggested that the change in TX90p was ahead of the change in runoff by 2 years.

### 5.2.3. Phase Relation between Runoff and ERI

The PRCPTOT–Q effect was relatively weak. The high energy vibration that mainly appeared at the 6-year time scale was concentrated in 1995–2005, but the correlation was not significant. The mutation year of PRCPTOT in 1995 represented the beginning year of high energy vibration, and its influence on low energy vibration at the 6-year time scale was apparent. The phase correlation between high energy vibration and low energy vibration was positive, and the phase angle was approximately 30°. This finding indicated that PRCPTOT had a lag effect on runoff changes, with a lag time of approximately 1 year. The runoff change before 1990 was mainly affected by the early rainfall in 1.5–2 years.

R10 and Q showed a significant relationship with the same phases and positive correlations between high and low energy vibrations at 1- to 2-year time scales in 2010–2015 and at a 6-year time scale in 2000–2010. The results indicated that R10 lagged behind runoff variation by 1.5 years at 1-, 2- and 6-year scales. R10 was mainly affected by rainfall from the previous 2 years before 2005–2010.

No significantly correlated region was observed for the effective spectral value of R17.3–Q high energy vibration region. Obvious high energy vibration zones and extremely significant low energy vibration zones had appeared since 1992. The low energy vibration zones at 5- to 7-year time scales from 1996 to 2007 passed the red-noise standard spectral test at the 0.05 significance level, showing a positive correlation in the same phase. The phase angle was approximately 30° at the 4-year time scale, 0° at the 5-year time scale, and 30° at 5- to 7-year time scales. On the basis of the results, the significant correlation in the low energy time–frequency region represented the impact of R17.3 on runoff change decreasing with time scale, from leading to consistent and then to lagging. The phase angle at 1- to 2-year time scales in 1992–2000 was approximately 30° in the effective spectrum value of the wavelet-affecting vertebrae. This finding indicated that R17.3 was 1 year ahead of the effect on runoff change at 1- to 2-year time scales. Moreover, obvious differences could be derived for the influence of R17.3 on runoff change at different time scales, and they mainly represented the low energy vibration effect.

The time–frequency relationship in the high energy zone of R20–Q was not significant, but it was significant in the low energy zone. It was found that low energy oscillation regions appeared at 4- to 7-year time scales, and all passed the red-noise standard spectral tests at the 0.05 significance level. The low energy oscillation region cantered in 1992 showed a negative correlation in the inverse phase with a phase angle of approximately 60°, indicating that the variation of R20 lagged behind that of runoff change by approximately 2 years. The low energy vibration region cantered in 2005 showed a positive correlation in the same phase with the phase angle ranging from 30° to 60°. The phase angle decreased with the year and increased with the time scale. The results showed that the lag time increased with time scales by approximately 1–2 years, and the variation of R20 and runoff tended to be consistent for a specific year. However, before the 2000s, runoff change at high time–frequency scales was affected by early rainfall for 1–2 years.

The significant correlation of R95p–Q vibration was weak, and no significantly correlated high energy time–frequency region was observed, whereas the significantly correlated low energy time–frequency region was not obvious. Only the regions at 4- to 5-year time scales from 1990 to 1995 passed the red-noise standard spectral test at the 0.05 significance level, and the positive correlation was depicted within the same phase. The phase angle increased from 0° in the initial year to 60° at this time scale, which indicated that the variation of R95p lagged behind the variation of runoff change by 0–3 years. It showed that a relatively obvious high and positive correlation region with the phase angle of 60° was observed at a 1-year time scale from 2004 to 2008, which indicated that the change in R95p lagged behind the change in runoff by approximately 3 years. Before the mid-2000s, the runoff change in high time–frequency was affected by early rainfall for 1–2 years.

The R99p–Q correlation in the high and low energy time–frequency regions did not pass the red-noise standard spectral test at the 0.05 significant level. According to the results of XWT, the correlation showed the positive correlations with different time–frequency effects at 4- and 6-year time scales. It exhibited a leading effect at the 4-year time scale, but a lagging effect at the 6-year time scale. After the 2000s, the runoff in the high time–frequency was affected by early rainfall for 1–2 years.

The Rx1day–Q correlation appeared in the significantly correlated region with an effective spectrum value and showed an inverse correlation at 5- to 6-year time scales from 2000 to 2010. The phase angle increased from 0° to 90° with the decrease of time scale. Rx1day lagged behind runoff change by approximately 3 years in the significantly correlated region, as shown by the positive correlation at the 4-year scale. In addition, the two changes had essentially the same characteristics. According to the results of WTC, the positive correlation appeared around 2005 at the 6-year time scale, and the phase angle gradually decreased along with the effect increases. The correlation area from 1990 to 2000 was much more obvious at the 4-year time scale, which indicated a positive correlation in the same phase. The phase angle gradually increased from 0° to 60°, as shown by the leading effect of 0–2 years.

The high energy time–frequency region of the Rx5day–Q correlation tested by XWT did not pass the red-noise standard spectral test at the 0.05 significance level. Within the effective spectral value of WTC, a local correlation area appeared at the 4-year time scale before and after 1998. However, the phase relationship of Rx5day–Q manifested a positive correlation with the phase angle of 45°, which indicated that the change in Rx5day lagged behind the change in runoff by 1.5 years. The invalidated spectral correlation value was highly significant outside the region after 2010. The Rx5day change was ahead of the runoff change by approximately 1.5 years according to phase identification, but the leading effect showed a gradual weakening trend. After the 2000s, the runoff variation at the 4-year time scale was mainly affected by rainfall in the previous 2 years, and it occurred mainly in the common high energy time–frequency region.

The results of XWT showed that the SDII–Q correlation appeared in the high energy oscillation region at the 6-year time scale before and after 2005 with an effective spectrum value. From a phase perspective, the phase angle that appeared at the same stage gradually increased to 45° over time, indicating that the impact of SDII on runoff was gradually lagging. According to the results of WTC (Figure 5), the high correlation region of low energy time–frequency zones was mainly concentrated at 4- to 8-year time scales after 1990, and passed the red-noise standard spectral test at the 0.05 significance level. From the phase perspective, the phase angle was reduced from 60° to 45° and had a positive correlation in the same phase, which indicated that the time lag of SDII behind that of runoff had gradually decreased and that the correlation between these two variables was enhanced. In addition, the characteristics of their changes at the 4-year time scale were relatively consistent and showed a positive correlation. In the high time–frequency scale before the 2000s, runoff change was mainly affected by early rainfall for 1–2 years.

In summary, the EPIs of R10 and Rx1day and the ETIs of SU34.4 and TNx were the main driving factors of runoff changes, which showed relatively large impacts on runoff

changes in high and low energy vibration regions. The changes of PRCPTOT, R17.3, R20, R95p, R99p, Rx5day, TX90p, and SDII had relatively large impacts on runoff changes only in low energy vibration regions. Overall, extreme precipitation events were ahead of runoff changes at the high time–frequency scale and exhibited a small lag effect at the low time–frequency scale, while extreme temperature events were mainly ahead of runoff changes.

## 6. Discussions

### 6.1. Influence of Abrupt Change for ECIs on Runoff Change

The high energy level of PRCPTOT, which was significantly higher than that for the surrounding time periods and time scales, was observed in 1994. A peak value appeared in the high energy time–frequency region in 2004, but it was not significant at the 0.05 confidence level. The significant interaction of R10, Rx1day, and SDII with runoff started to appear in the high energy time–frequency region around 2003, which was the abrupt change year of runoff change in the Yinjiang River watershed. Rainfall had abruptly changed in 1994 and 2004 [10], and the intensity of the interaction between rainfall and runoff had also changed accordingly. This finding indicated that runoff was greatly affected by rainfall.

This study further detected the high energy time–frequency region and the significantly correlated region of the interaction between ECIs and runoff change before and after the abrupt change (Table 5). The time of ECIs leading and lagging runoff changes was affected by the abrupt time. The vibration-related regions of the interaction between ECIs and runoff shifted between high and low time–frequency. Abrupt change transformed the scale of the region where ECIs interacted with runoff, which then led to abrupt change in runoff.

**Table 5.** Change trend and abrupt change times of ECIs calculated by Mann–Kendall method.

| ECIs | Change Trend | Abrupt Change Time | Variation Time of High Energy and Significant Correlation Regions | |
|---|---|---|---|---|
| | | | XWT | WTC |
| PRCPTOT | + | 1994 2010 | 2009 | 2009 |
| R10 | + | 1993 2004 2013 | 2001 2010 | 1995 2013 |
| R17.3 | + | 1992 2008 | 2011 | 1996 2007 |
| R20 | + | 1992 2004 | - | 1993 2002 2008 |
| R95p | − | 1994 2002 2013 | - | 1993 |
| R99p | − | 1997 2004 2013 | - | - |
| Rx1day | − | 1990 2004 2013 | 2001 2009 | 1993 2013 |
| Rx5day | + | 1987 2013 | 2011 | 1993 2002 |
| SDII | − | 1994 2004 | 2005 | 1993 2013 |
| SU34.4 | + | 2004 2013 | 2006 | 2000 2010 |
| TNx | + | 1991 2004 | 2006 2010 | 2000 2011 |
| TX90p | + | 2001 | 2003 | - |

In addition, SU34.4 and TNx were significant in high energy time–frequency regions after 2004. The high energy time–frequency regions of TX90p also appeared in this period, and the values were significantly higher than those of the surrounding periods and time scales. The year 2004 represented the sudden-change year of SU34.4 and TNx, whilst 2005 represented the sudden-change year of temperature. After 2006, the effect of temperature on runoff changed significantly and showed an advanced effect.

### 6.2. Influence of Temperature and Precipitation on Runoff Change

The findings fully demonstrated that extreme climate change was the main driving factor of the runoff increase, and the transition of the interaction between ECIs and runoff in high and low frequency scales was related to the catastrophic characteristics of ECI. The energy of the high energy time–frequency region in which R10 and Rx1day interacting with runoff was high for the significantly extreme precipitation, and the range of the significantly

correlated region was relatively large. The influence of R17.3 and R20 was relatively small, mainly under the action of low energy vibration. In terms of the ETIs, SU34.4 and TNx had a relatively large impact on runoff change. The variation characteristics of extreme precipitation were obviously more intense than those of ETI. However, the issue of which had the greater impact—extreme precipitation or extreme temperature—on runoff change had been the subject of little research. The previous studies mainly focused on climate change (e.g., rainfall, evaporation, and temperature) and the impact of human activities on runoff change, but their results were quite different.

The previous studies had also shown that rainfall and temperature played significant roles in runoff change. Variations in temperature and precipitation had significant effects on runoff in the four headstreams of the Tarim River. In particular, precipitation had a positive impact on water flow in Aksu River, Hotan River, and Kaidu River, whilst temperature had a positive impact on water flow in Yarkant River [54].

However, other studies argued that the main cause of runoff change was rainfall rather than temperature. A correlation existed between rainfall change and runoff change, as depicted by a linear multiple regression equation, which also indicated that changes in annual runoff were contributed to by the changes in annual PT rather than by changes in air temperature [45]. The effects of recent temperature fluctuations on stream flow were minimal, but the impact of relatively small fluctuations in precipitation (approximately 10%) was often amplified by two or more factors and depending on basin and climate characteristics [55]. Model outputs were used to examine the separate effects of total precipitation and temperature on runoff variability in the conterminous United States, and modelling results showed that water–year runoff had increased in this region, in which precipitation accounted for nearly all water–year runoff variability in the past century. By contrast, temperature effects on runoff had been negligible for most locations in the United States, even during periods when temperatures increased significantly. The extant watershed research showed that runoff and precipitation were well-synchronized with abrupt change features and stage characteristics, and they exhibited consistent multi-timescale characteristics. The contribution of precipitation to runoff change was 50%–60% and was considered high but stable in the Yinjiang River watershed [10,11]. Meanwhile, the extant karst basin research showed that changes in water discharge were mainly controlled by variations in precipitation [56]. The detected trends in water discharge had been mainly attributed to the climate change in the eastern and north-western Pacific sides of the United States [57,58]. Climate oscillations could cause precipitation changes that further affected annual variations in water discharge [47,59]. Evidently, the correlations between annual regional precipitation and water discharge were all significant, which implied that total precipitation exerted substantial influence on water discharge [60,61]. All these previous studies indicated that rainfall was an important factor affecting runoff change. However, some studies had presented contrasting results. For instance, runoff in non-karst areas was reportedly more sensitive to temperature, which might be an important factor affecting rainfall. A SWAT model was used to analyse the impacts of climate change and land cover on runoff and predict future runoff trends in river basins [62], and finally found that runoff was more sensitive to temperature than precipitation.

### 6.3. Response Mechanism of Runoff Change to Different ECIs

The effects of different ECIs on runoff change were markedly different. The energy of the high energy time–frequency region, in which R10 and Rx1day interacted with runoff during significantly extreme precipitation, was higher than that of other EPIs, and the range of the significantly correlated region was relatively large. For the ETIs, the effects of SU34.4 and TNx on runoff change were relatively larger. R10 occurred frequently and intensively at a maximum of 44 days/year with a minimum of 18 days/year, and each rainfall was greater than 10 mm. The maximum of Rx1day was 151.5 mm/day with a minimum of 39.8 mm/day. Given its distinctive lithological background, surface runoff was unlikely to occur in the condition with a single weak rainfall due to the highly fragmented surface, exposed and prominent bedrock, and low runoff coefficient in karst area. Moreover, most

of the rainfall was transformed into underground runoff through karst fractures during rainfall, which rendered it difficult for weak rainfall to affect surface runoff. The frequency of R10 was very high compared with the others, and the corresponding rainfall intensity was larger. As such, a cumulative fine path flow could be generated. The maximum daily rainfall in most years could be directly attributed to Rx1day. The effects of R10 and Rx1day on runoff change were direct. Meanwhile, the effects of R17.3 and R20 on runoff change were relatively small, and the main effect could be represented by a single rainfall that produced slope runoff that affected runoff change under low energy vibrations. The frequencies of R17.3 and R20 were lesser than those of R10 and, thus, their effects on runoff change were also smaller. This finding could be explained by the presence of relatively scarce karst surface water and relatively small soil moisture covered on the karst basement [63]. Furthermore, the instantaneous slope runoff formed by rainfall was initially needed to satisfy soil moisture saturation requirements. This resulted in the initial rainfall being first absorbed by the soil, preventing timely formation of river runoff.

Rainfall and temperature represented available water and energy in basins, which in turn controlled the distribution of runoff and evaporation [64,65]. Runoff change was strongly affected by changes in evaporation [10,11]. Temperature changes the effects of evaporation and rainfall intensity on runoff. The previous studies had shown that the water cycle was mainly controlled by transpiration in vegetation-developed karst watersheds, suggesting that evaporation plays an important role in runoff variation [66–69]. According to the direction of phase angles, SU34.4 and runoff change represented a negative phase, as shown by their negative correlation. This finding indicated that the significant increase of SU34.4 increased evaporation, and it exhibited a negative effect on runoff change. TNx was positively correlated with runoff change, indicating that the increase of TNx might increase rainfall to produce marked effects on runoff change. Therefore, changes in ETIs, including sudden changes, did not necessarily have a direct impact on runoff changes, but indirectly affected runoff by changing the frequency and intensity of rainfall and evaporation.

*6.4. Limitations and Future Prospects*

This study focused on the impact of extreme climatic events on runoff variation. The extreme climatic events, which were significantly related to runoff variation, were qualitatively discussed at an inter-annual scale, and their correlation with temporal variability of runoff variation was presented. However, the study only indirectly presented the comprehensive effects of these impacts through the intensity of rainfall and evaporation, but it did not quantify the impact of specific extreme events, which might not be sufficient for showing the magnitude of the effects of certain extreme climatic events on runoff changes. In the future, it will be necessary to quantitatively evaluate the impact and contribution of extreme climatic events on and to runoff changes, further predict potential hydrological processes and hazards that may occur in the future, and provide decision-making support for hydrological regulation and water resource security management in karst basins.

**7. Conclusions**

In this study, the multi-scale influences of extreme climatic events on runoff changes in the Yinjiang River watershed were identified by using wavelet analysis based on a 32-year hydrometeorological time series, and time–frequency response relationships between runoff changes and ECIs with significant impact were further revealed at different time scales. The main results were as follows:

(1) Twelve ECIs were detected to have a significant impact on runoff changes, including three ETIs (SU34.4, TNx, and TX90p) and nine EPIs (SDII, PRCPTOT, R10, R17.3, R20, R95p, R99p, Rx1days, and Rx5days), mainly at a 6-year time scale.

(2) The R10 and Rx1day in ERI and SU34.4 and TNx in ETI were the main driving factors for recent runoff changes, which exhibited relatively large impacts on runoff changes in high and low energy vibration regions. However, the remaining eight ECIs that passed

the 0.05 confidence level showed relatively large impacts on runoff changes only in low energy vibration regions.

(3) The transition of the interaction between ECIs and runoff changes in high and low time–frequency scales was related to the abrupt change characteristics of the ECIs. The correlation of abrupt change was an important reason for the emergence of highly correlated regions that trigger high and low energy vibrations.

(4) As a whole, the extreme precipitation events were ahead of runoff changes at the high frequency scale and exhibited small lag effects at the low frequency scale, while extreme temperature events were mainly ahead of runoff changes.

**Author Contributions:** Conceptualization, methodology and validation, L.W.; Software and data curation, L.W.; Formal analysis, L.W., D.Y. and D.C.; Investigation, G.L. and J.W.; Writing—original draft preparation, L.W. and F.C. All authors have read and agreed to the published version of the manuscript.

**Funding:** This research work was supported jointly by the National Natural Science Foundation of China (No. 42261052); Guizhou Provincial Science and Technology Projects (No. ZK [2023]-464); Science and Technology Projects of Tongren City (No. 2022-63); Scientific Research Projects in Higher Education Institutions of Guizhou Provincial Department of Education (Youth Project) (No. 2022-351); Doctoral Research Startup Fund Project of Tongren University (No. trxyDH2103); Guizhou Provincial Major Science and Technology Achievement Transformation Project (No. 2022-010); and the Opening Fund of the State Key Laboratory of Environmental Geochemistry (No. SKLEG2021072001).

**Institutional Review Board Statement:** Not applicable.

**Informed Consent Statement:** Not applicable.

**Data Availability Statement:** The data analyzed in this study are subject to the following licenses/ restrictions: The dataset can only be accessed from China Meteorological Data Sharing Service System, Karst Scientific Data Center and Guizhou Provincial Hydrology and Water Resources Bureau. Requests to access these datasets should be directed to jgywlh@gztrc.edu.cn.

**Acknowledgments:** We would like to thank all the authors and reviewers for their significant guidance and help in writing this manuscript.

**Conflicts of Interest:** The authors declare no conflict of interest.

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
