# Peer review of "Response of Runoff Change to Extreme Climate Evolution in a Typical Watershed of Karst Trough Valley, SW China"

_atmosphere, doi:10.3390/atmos14060927_

Round 1

Reviewer 1 Report

The objective of this paper is to diagnose extreme climatic events with significant impacts on runoff changes in typical watershed of karst trough valley areas, identify the response characteristics of runoff changes to extreme temperature events and clarify the coupling relationships, phase relations and temporal variability between significantly extreme climatic events and runoffs in common high- and low-energy regions in different time scales. Moreover, theoretical basis and scientific and technological support for the safety management of water resources in karst watersheds area was also provided.

This is an interesting and well-structured paper. All necessary sections (Introduction, Study site, Materials, Methodology, Result Analysis, Discussions, Conclusions) have been considered. Moreover, the “Methodology”, “Result Analysis” and “Discussions” sections are divided into sub-sections, providing additional details. Furthermore, all Figures, Tables and Diagrams are consistent with the analysis provided in the manuscript. Regarding the mathematical part, predominantly analyzed in the “Methodology” section, it is valid and satisfactorily explained. However, some changes should be implemented, which will improve the paper. In particular:

Lines 13-38: The paper abstract should be briefer and more concise. All additional details should be placed in the main body of the paper. Please, apply.

Lines 62-64: This part lacks bibliographic references, related to runoff in karst watersheds. Additional information can also be obtained from the following papers, which can optionally be cited: 1. Banks, D., Odling, N.E., Skarphagen, H., Rohr-Torp, E., 1996. Permeability and stress in crystalline rocks. Terra Nov. 8, 223–235. https://doi.org/10.1111/j.1365-3121.1996.tb00751.x, 2. Manoutsoglou, E., Lazos, I., Steiakakis, E., Vafeidis, A., 2022. The Geomorphological and Geological Structure of the Samaria Gorge, Crete, Greece—Geological Models Comprehensive Review and the Link with the Geomorphological Evolution. Appl. Sci. 12, 10670. https://doi.org/10.3390/app122010670, 3. Lipar, M., Ferk, M., 2015. Karst pocket valleys and their implications on Pliocene–Quaternary hydrology and climate: Examples from the Nullarbor Plain, southern Australia. Earth-Science Rev. 150, 1-13. https://doi.org/10.1016/j.earscirev.2015.07.002. Please, apply.

Lines 161-162: Please, describe the major geological structures (e.g. faults, thrusts etc.) and lithological formations of the study area. A geological map should be added, including tectonic and lithological features. Please, apply.

Line 176: Please, provide a more detailed description in the Figure 1 caption.

Lines 352-353: Please, provide Figure 4 in a higher resolution. It contains blurry parts in the current form. Moreover, improve the description in the corresponding caption, please.

Lines 396-397: Please, provide Figure 5 in a higher resolution. In addtion, improve the description in the corresponding caption, please.

Lines 576-602: The “Conclusions” section should be modified. In its current form, the main results are highlighted, as mentioned in the Line 580. This section should be comprehensive, while the major findings of the paper should be highlighted. Maybe, numbering of the conclusion remarks could be performed. Please, apply.

Author Response

Dear Editors and Reviewers:

Thank you and reviewers for the extremely helpful comments concerning our manuscript entitled “Responses of runoff changes to extreme climate evolution in a typical watershed of a karst trough valley, SW China” (atmosphere-2399900). Those comments are all valuable and very helpful for revising and improving our paper, as well as the important guiding significance to the researches. Thank you for your patience and the reviewers for their valuable comments and advice. We have studied comments carefully and tried our best to revise the manuscript according to every single comment which made by the reviewers and the editors. We have marked every changes with highlight colors in this revised version (See the the updated “Manuscript (revision changes marked)).

Point-by-point responses to the reviewers’ comments are listed below this letter. According to the editors and reviewers' suggestions, we have systematically revised our Manuscript. We hope that the revised version of the manuscript is now acceptable for publication in your journal.

Thank you for your consideration!

Sincerely yours,

Luhua Wu

*Corresponding Author: Luhua Wu

====================================================Point-by-Point Responses to Editor and Reviewers Comments

====================================================

Thank you for the patience and careful work. Both the technical and grammatical

revisions have been made thoroughly in the updated PDF version of “Manuscript (revision changes marked)”. The point-by-point responses to the reviewer’s comments are listed as following:

Response to Reviewer 1 Comments

Comments: The objective of this paper is to diagnose extreme climatic events with significant impacts on runoff changes in typical watershed of karst trough valley areas, identify the response characteristics of runoff changes to extreme temperature events and clarify the coupling relationships, phase relations and temporal variability between significantly extreme climatic events and runoffs in common high- and low-energy regions in different time scales. Moreover, theoretical basis and scientific and technological support for the safety management of water resources in karst watersheds area was also provided.

This is an interesting and well-structured paper. All necessary sections (Introduction, Study site, Materials, Methodology, Result Analysis, Discussions, Conclusions) have been considered. Moreover, the “Methodology”, “Result Analysis” and “Discussions” sections are divided into sub-sections, providing additional details. Furthermore, all Figures, Tables and Diagrams are consistent with the analysis provided in the manuscript. Regarding the mathematical part, predominantly analyzed in the “Methodology” section, it is valid and satisfactorily explained. However, some changes should be implemented, which will improve the paper.

Overall Response: Thank you very much for your significant comments and instructive advice, which is really important for us to modify and improve the manuscript and has significant guiding significance for our future work. Your valuable advice points out the direction for our future research. According to your nice suggestions, we have tried our best to improve the manuscript and made some revises in the manuscript. Our point-by-point responses are marked in blue font. We have marked every change with highlight color in the updated PDF version of “Manuscript (revision changes marked)”. For the all questions you mentioned, my responses and improvements are as follows.

Point-by-Point Responses to Each Question of Reviewer

Questions 1: Lines 13-38: The paper abstract should be briefer and more concise. All additional details should be placed in the main body of the paper. Please, apply.

Response: Thank you for your valuable suggestions. We have already tried to keep the Abstract as concise as possible according to your nice suggestions. Moreover, we have also checked all the language errors carefully. Please see on line 13-34 of the “Abstract in the updated PDF version of “Manuscript (revision changes marked)”.

Questions 2: Lines 62-64: This part lacks bibliographic references, related to runoff in karst watersheds. Additional information can also be obtained from the following papers, which can optionally be cited: 1. Banks, D., Odling, N.E., Skarphagen, H., Rohr-Torp, E., 1996. Permeability and stress in crystalline rocks. Terra Nov. 8, 223–235. https://doi.org/10.1111/j.1365-3121.1996.tb00751.x, 2. Manoutsoglou, E., Lazos, I., Steiakakis, E., Vafeidis, A., 2022. The Geomorphological and Geological Structure of the Samaria Gorge, Crete, Greece—Geological Models Comprehensive Review and the Link with the Geomorphological Evolution. Appl. Sci. 12, 10670. https://doi.org/10.3390/app122010670, 3. Lipar, M., Ferk, M., 2015. Karst pocket valleys and their implications on Pliocene–Quaternary hydrology and climate: Examples from the Nullarbor Plain, southern Australia. Earth-Science Rev. 150, 1-13. https://doi.org/10.1016/j.earscirev.2015.07.002. Please, apply.

Response: Thank you very much for your instructive advice. We have supplemented the theoretical knowledge related to karst watershed runoff in this section and cited the suitable references you recommended. Thank you for your recommendation of these articles. After reading them, we have learned more in-depth knowledge of karst theory. 

The supplementary literature is as follows:

  1. Min,R.Y.; Gu,X.H.; Yansong Guan,Y.S.; Zhang, X. Increasing likelihood of global compound hot-dry extremes from temperature and runoff during the past 120 years. J. Hydrol. 2023, 621,129553.
  2. Labat, D.;Goddéris, Y.; Probst, J. L.; Guyot, J. L. Evidence for global runoff increase related to climate warming. Adv. Water Resour. 2004, 27(6):631-642.
  3. Zscheischler, J.; Westra, S.; van den Hurk, B.J. J. M.; Seneviratne, S.I.; Ward, P. J.; Pitman, A.; et al. Future climate risk from compound events. Nat. Clim. Change. 2018, 8, 469-477.
  4. Hao, Z.;Hao, F.; Singh, V. P.;Ouyang, W. Quantitative risk assessment of the effects of drought on extreme temperature in eastern China. J Geophys. Res-Atmos. 2017, 12(17):9050-9059.
  5. Banks, D.; Odling, N.E.; Skarphagen, H.; Rohr-Torp, E. Permeability and stress in crystalline rocks. Terra Nov. 1996, 8:223–235.
  6. Manoutsoglou, E.; Lazos, I.; Steiakakis, E.; Vafeidis, A. The geomorphological and geological structure of the Samaria Gorge, Crete, Greece—geological models comprehensive review and the link with the geomorphological evolution. Appl. Sci. 2022, 12, 10670.
  7. Lipar, M.; Ferk, M. Karst pocket valleys and their implications on Pliocene–Quaternary hydrology and climate: Examples from the Nullarbor Plain, southern Australia. Earth-Science Rev. 2015, 150, 1-13.

Please see on line 60-66 in the updated PDF version of “Manuscript (revision changes marked)”.

Questions 3: Lines 161-162: Please, describe the major geological structures (e.g. faults, thrusts etc.) and lithological formations of the study area. A geological map should be added, including tectonic and lithological features. Please, apply.

Response: Thank you for your comment. We have added a description of the main geological structures and lithological strata in the study area, as well as a lithological map. The supplementary content is as follows: “The basin was jointly constrained by 8 geological backgrounds, and its surface topography was controlled by 6 lithological coupling factors. The karst area in the Yinjiang River basin was mainly controlled by Homogenous dolomite, Clastic rock of limestone interlayer, and Interbedded limestone and clastic rock (Figure. 1c), with controlled areas of 108.29 km2, 106.81 km2, and 69.12 km2 respectively, accounting for 15.66%, 15.44%, and 9.99% of the total area (Table 1)”.

Please see on line 162-168 and line 193-194 in the updated PDF version of “Manuscript (revision changes marked)”.

Questions 4: Line 176: Please, provide a more detailed description in the Figure 1 caption.

Response: Thank you for your comment. We have already tried to keep it as concise as possible according to your nice suggestions. We have provided a more detailed description in the Figure 1 caption. Since the detailed information in the figure has already been described in the paragraph, we have only improved the title of Figure 1. The improved description in the Figure 1 caption is as follows: “The location of study area in China (a), DEM (b), and lithology (c) in the Yinjiang River watershed.”. Please see on line 155-156 in the updated PDF version of “Manuscript (revision changes marked)”.

Questions 5:Lines 352-353: Please, provide Figure 4 in a higher resolution. It contains blurry parts in the current form. Moreover, improve the description in the corresponding caption, please.

Response: Thank you for your comment. We have provided Figure 4 in a higher resolution and improved the description in the corresponding caption. The improved the description is as follows: “Figure 4. The wavelet power spectrum for ECIs and runoff (Q) in the Yinjiang River watershed from 1984 to 2015. The thick black contour designated the 5% significance level for red noise, and the cone of influence (COI) where edge effects might distort the picture was shown as a lighter shade. Phase change reflected the difference in response time of primary variables to influence factors. The phase relationship between influence factors and primary variables was indicated by arrows. The arrows from left to right indicated that the influencing factors and primary variables were in the same phase, which implied a positive correlation; the arrows from right to left indicated an inverse phase, which implied a negative correlation; the downward arrows indicated that influencing factors were 90â—¦ ahead of primary variables, and the upward arrows indicated that influence factors were 90â—¦ behind primary variables.”. Please see on line 363-374 in the updated PDF version of “Manuscript (revision changes marked)”.

Questions 6: Lines 396-397: Please, provide Figure 5 in a higher resolution. In addtion, improve the description in the corresponding caption, please.

Response: Thank you for your comment. We have provided Figure 5 in a higher resolution and improved the description in the corresponding caption. The improved the description is as follows: “Figure 5. The wavelet condensation spectrum for ECIs and runoff (Q) in the Yinjiang River watershed from 1984 to 2015. The thick black contour designated the 5% significance level for red noise, and the cone of influence (COI) where edge effects might distort the picture was shown as a lighter shade. Phase change reflected the difference in response time of primary variables to influence factors. The phase relationship between influence factors and primary variables was indicated by arrows. The arrows from left to right indicated that the influencing factors and primary variables were in the same phase, which implied a positive correlation; the arrows from right to left indicated an inverse phase, which implied a negative correlation; the downward arrows indicated that influencing factors were 90â—¦ ahead of primary variables, and the upward arrows indicated that influence factors were 90â—¦ behind primary variables.”. Please see on line 418-429 in the updated PDF version of “Manuscript (revision changes marked)”.

Questions 7: Lines 576-602: The “Conclusions” section should be modified. In its current form, the main results are highlighted, as mentioned in the Line 580. This section should be comprehensive, while the major findings of the paper should be highlighted. Maybe, numbering of the conclusion remarks could be performed. Please, apply.

Response: Thank you for your comment. We have modified the “Conclusions” section and numbered the conclusion remarks. We briefly list the most important findings in the conclusions. Please see on line 608-628 in the updated PDF version of “Manuscript (revision changes marked)”.

Reviewer 2 Report

This MS studies the evolution relationship and time-frequency effects of runoff and extreme climate change in typical karst valley basins at different time scales through continuous wavelet transform (CWT), cross wavelet transform (XWT), and wavelet coherence transform (WTC). In my opinion, it is an interesting study and elucidates the multi temporal scale impact mechanism of runoff changes and extreme climate evolution in hydrological processes in karst basins.

Some minor comments as below.

Line 17: “are”, change it to “were”.

Line 19: , change “Show” to “showed”, change “are” to “were”.

Line 22: “are”, change it to “were”.

Line 24: “have”, change it to “had”.

Line 24: change ”is” to “was”.

Line 26-27: Rewrite this sentence ”This abrupt change is the most important factor of runoff changes, and it serves as basis for the emergence of high-energy zones and height-related  regions with low energy.”

Line 28: Delete this meaningless sentence “In addition, different significant ECIs have different effects on runoff change in different years and time scales. ”

Line 29-30: Rewrite this sentence “As a whole, extreme rainfall events are ahead of runoff change in high frequency scale, and lag effect in low frequency scale is small, while extreme temperature events are mainly ahead of runoff change. “

Line 103: “extreme climate indices” It is the first time it appears and should be marked with its abbreviated form.

Line 103: change “are” to “were”.

Line 103: “is a typical karst valley watershed in a typical watershed of a karst trough valley”. This sentence is not grammatical. The meaning expressed is very confusing.

Line 194: change ”is” to “was”.

Line 214: “0.349 and 0.449“. It can be retained 2 decimal places.

Line 218: Change “indicate” to “indicated”.

Line 219: Change “have” to “had”, Change “can” to “could”.

Line 221: Change “represent” to “represents”.

Line 241: Change “appear” to “appeared ”.

Line 242-246: change ”is” to “was”, change ”vary” to “varied”, Change “appear” to “appeared ”.

Line 247: change ”change” to “changed ”.

Line 248-250: Suggest using the past tense in sentences.

Line 277-283: change ”is” to “was”, Change “appears” to “appeared ”.

Line 321-323 and line 355-357: Rewrite these sentence.

The runoff mutation time expressed in Table 4 in line 366-367 already includes the detected mutation time in Figure 6. It is recommended to delete Figure 6 in Line 371-372.

Line 471-475: Too verbose, please optimize this conclusion.

The manuscript is written in a good way, however, there is a need to improve the introduction. The author(s) could describe the problem in a better way.

Suggest using the past tense for the description of the result analysis.

Please check the paper carefully for language errors.

You should underline the practical value of your study in relation to the water management problems faced by Guizhou Province. Not just karst basins.

Author Response

Dear Editors and Reviewers:

First of all, thank you and reviewers for the extremely helpful comments concerning our manuscript entitled “Responses of runoff changes to extreme climate evolution in a typical watershed of a karst trough valley, SW China” (atmosphere-2399900). Those comments are all valuable and very helpful for revising and improving our paper, as well as the important guiding significance to the researches. Thank you for your patience and the reviewers for their valuable comments and advice. We have studied comments carefully and tried our best to revise the manuscript according to every single comment which made by the reviewers and the editors. We have marked every changes with highlight colors in this revised version (See the the updated “Manuscript (revision changes marked)).

Point-by-point responses to the reviewers’ comments are listed below this letter. According to the editors and reviewers' suggestions, we have systematically revised our Manuscript. We hope that the revised version of the manuscript is now acceptable for publication in your journal.

Thank you for your consideration!

Sincerely yours,

Luhua Wu

*Corresponding Author: Luhua Wu

====================================================Point-by-Point Responses to Editor and Reviewers Comments

====================================================

Thank you for the patience and careful work. Both the technical and grammatical

revisions have been made thoroughly in the updated PDF version of “Manuscript (revision changes marked)”. The point-by-point responses to the reviewer’s comments are listed as following:

Response to Reviewer 2 Comments

Comment: This MS studies the evolution relationship and time-frequency effects of runoff and extreme climate change in typical karst valley basins at different time scales through continuous wavelet transform (CWT), cross wavelet transform (XWT), and wavelet coherence transform (WTC). In my opinion, it is an interesting study and elucidates the multi temporal scale impact mechanism of runoff changes and extreme climate evolution in hydrological processes in karst basins.

Some minor comments as below. 

Overall Response: Thanks for your professional review and valuable advice on our paper. We feel sincerely thanks for your agreement with our paper. As you are concerned, there are several problems that need to be addressed. According to your nice suggestions, We will be happy to edit the text further, based on helpful comments from the reviewers. And we will try our best to improve the manuscript and make detailed revises in the manuscript. Our point-by-point responses are marked in blue font, and the parts is marked in bold. We have marked every change with highlight color in the updated PDF version of “Manuscript (revision changes marked)”. We appreciate for your warm work earnestly, and hope that the correction will meet with approval. The detailed corrections are listed below.

Questions 1: Line 17: “are”, change it to “were”.

Response: Thank you for your valuable suggestions. We have corrected it. The modified marks can be seen on line 19 in the updated PDF version of “Manuscript (revision changes marked)”.

Questions 2: Line 19: change “Show” to “showed”, change “are” to “were”.

Response: Thank you for your valuable suggestions. We have revised this sentence. The modified marks can be seen on line 21-22 in the updated PDF version of “Manuscript (revision changes marked)”.

Questions 3: Line 22: “are”, change it to “were”.

Response: Thank you for your valuable suggestions. We have corrected it. The modified marks can be seen on line 23 in the updated PDF version of “Manuscript (revision changes marked)”.

.Questions 4: Line 24: “have”, change it to “had”. 

Response: Thank you for your valuable suggestions. We have revised this sentence. . The modified marks can be seen on line 25-26 in the updated PDF version of “Manuscript (revision changes marked)”.

Questions 5: Line 26: change ”is” to “was”.

Response: Thank you for your valuable suggestions. We have corrected it. The modified marks can be seen on line 27 in the updated PDF version of “Manuscript (revision changes marked)”.

Questions 6: Line 26-27: Rewrite this sentence ”This abrupt change is the most important factor of runoff changes, and it serves as basis for the emergence of high-energy zones and height-related  regions with low energy.”

Response: Thank you for your valuable suggestions. We have rewritten this sentence. The rewritten sentence is: “The correlation of abrupt change was an important reason for the emergence of highly correlated regions that trigger high and low energy vibrations”. The modified marks can be seen on line 28-29 in the updated PDF version of “Manuscript (revision changes marked)”.

Questions 7: Line 28: Delete this meaningless sentence “In addition, different significant ECIs have different effects on runoff change in different years and time scales. ”

Response: Thank you for your valuable suggestions. We have deleted this sentence. The modified marks can be seen on line 30-31 in the updated PDF version of “Manuscript (revision changes marked)”.

Questions 8: Line 29-30: Rewrite this sentence “As a whole, extreme rainfall events are ahead of runoff change in high frequency scale, and lag effect in low frequency scale is small, while extreme temperature events are mainly ahead of runoff change. “ 

Response: Thank you for your valuable suggestions. We have rewritten this sentence. The rewritten sentence is: “As a whole, the extreme precipitation events were ahead of runoff changes at high time-frequency scale and exhibited small lag effects at low time-frequency scale, while extreme temperature events mainly were ahead of runoff changes.”. The modified marks can be seen on line 30-32 in the updated PDF version of “Manuscript (revision changes marked)”.

Questions 9: Line 103: “extreme climate indices” It is the first time it appears and should be marked with its abbreviated form.

Response: Thank you for your valuable suggestions. We are very sorry, this is an expression error and we have corrected it. The modified marks can be seen on line 128 in the updated PDF version of “Manuscript (revision changes marked)”.

Questions 10: Line 103: change “are” to “were”.

Response: Thank you for your valuable suggestions. We have corrected it. The modified marks can be seen on line 128 in the updated PDF version of “Manuscript (revision changes marked)”.

Questions 11: Line 103: “is a typical karst valley watershed in a typical watershed of a karst trough valley”. This sentence is not grammatical. The meaning expressed is very confusing. 

Response: Thank you for your valuable suggestions. We have corrected it. The modified marks can be seen on line 124 in the updated PDF version of “Manuscript (revision changes marked)”.

Questions 12: Line 194: change ”is” to “was”.

Response: Thank you for your valuable suggestions. We have corrected it. The modified marks can be seen on line 246 in the updated PDF version of “Manuscript (revision changes marked)”.

Questions 13: Line 214: “0.349 and 0.449“. It can be retained 2 decimal places. 

Response: Thank you for your valuable suggestions. We have corrected it. The modified marks can be seen on line 271 in the updated PDF version of “Manuscript (revision changes marked)”.

Questions 14: Line 218: Change “indicate” to “indicated”.

Response: Thank you for your valuable suggestions. We have corrected it. The modified marks can be seen on line 277 in the updated PDF version of “Manuscript (revision changes marked)”.

Questions 15: Line 219: Change “have” to “had”, Change “can” to “could”.

Response: We agree. Thank you for your valuable suggestions. We have corrected it. The modified marks can be seen on line 277-278 in the updated PDF version of “Manuscript (revision changes marked)”.

Questions 16: Line 221: Change “represent” to “represents”.

Response: We agree. Thank you for your valuable suggestions. We have corrected it. The modified marks can be seen on line 280-281 in the updated PDF version of “Manuscript (revision changes marked)”.

Questions 17: Line 241: Change “appear” to “appeared ”.

Response: We agree. Thank you for your valuable suggestions. We have corrected it. The modified marks can be seen on line 308 in the updated PDF version of “Manuscript (revision changes marked)”.

Questions 18: Line 242-246: change ”is” to “was”, change ”vary” to “varied”, Change “appear” to “appeared ”.

Response: Thank you for your comment. We have corrected it. The modified marks can be seen on line 309-312 in the updated PDF version of “Manuscript (revision changes marked)”.

Questions 19:  Line 247: change ”change” to “changed ”.

Response: Thank you for your valuable suggestions. We have corrected it. The modified marks can be seen on line 315-316 in the updated PDF version of “Manuscript (revision changes marked)”.

Questions 20:  Line 248-250: Suggest using the past tense in sentences.

Response: Thank you for your comment. We have corrected it. The modified marks can be seen on line 317-320 in the updated PDF version of “Manuscript (revision changes marked)”.

Questions 21: Line 277-283: change ”is” to “was”, Change “appears” to “appeared ”.

Response: We agree. Thank you for your comment. We have corrected it. The modified marks can be seen on line 355-363 in the updated PDF version of “Manuscript (revision changes marked)”.

Questions 22: Line 321-323 and line 355-357: Rewrite these sentence.

Response: We agree. Thank you for your valuable suggestions. We have rewritten this sentence. The rewritten sentence for line 321-323 is: “According to the results of XWT, the correlation showed the positive correlations with different time-frequency effects at 4- and 6-year time scales. It exhibited a leading effect on the 4-year time scale, but a lagging effect on the 6-year time scale.”. The modified marks can be seen on line 432-434 in the updated PDF version of “Manuscript (revision changes marked)”.

The rewritten sentence for line 355-357 is: “Overall, extreme rainfall events were ahead of runoff changes at the high-frequency scale and exhibited small lag effects at the low-frequency scale, while extreme temperature events mainly were ahead of runoff changes.” The modified marks can be seen on line 476-479 in the updated PDF version of “Manuscript (revision changes marked)”.

Questions 23: The runoff mutation time expressed in Table 4 in line 366-367 already includes the detected mutation time in Figure 6. It is recommended to delete Figure 6 in Line 371-372.

Response: We agree. Thank you for your valuable suggestions. We strongly agree with your suggestion. We have deleted Figure 6 according to your nice suggestions. The modified marks can be seen on line 492-504 in the updated PDF version of “Manuscript (revision changes marked)”.

Questions 24: Line 471-475: Too verbose, please optimize this conclusion.

Response: Thats a good idea. Thank you for your valuable suggestions. We are grateful for the comment. The conclusion has been optimized now. The modified marks can be seen on line 608-628 in the updated PDF version of “Manuscript (revision changes marked)”.

Questions 25: The manuscript is written in a good way, however, there is a need to improve the introduction. The author(s) could describe the problem in a better way.

Response: Thank you for your patience and careful work. We have comprehensively revised the introduction in combination with your suggestions. Moreover, we have also checked all the language errors and various details carefully for the introduction according to your suggestions and requirements.. Please see on line 43-139 of the “Section 1. Introduction in the updated PDF version of “Manuscript (revision changes marked)”.

Questions 26: Suggest using the past tense for the description of the result analysis.

Response: Thank you for your patience and careful work.we have used the past tense for the description of the result analysis and corrected the unreasonable tense in the sentence. The modified marks can be seen in the updated PDF version of “Manuscript (revision changes marked)”.

Questions 27: Please check the paper carefully for language errors.

Response: Thank you for your patience and careful work. We apologized for the poor language of our manuscript. Your suggestions made us aware of the language errors in our manuscript. Now, we have checked all the language errors carefully for our Manuscript. Please see on revision mark in the updated “Manuscript (revision changes marked)”.

In addition, we have carefully checked and corrected the reference formats in the whole revised manuscript according to the requirements of the Atmosphere. For example, the revision of journal name abbreviation and verification of interval symbols were corrected carefully. Please see the section of ‘Reference’ in the updated “Manuscript (revision changes marked)”. 

Questions 28:You should underline the practical value of your study in relation to the water management problems faced by Guizhou Province. Not just karst basins.

Thank you for your valuable suggestions. We have emphasized the scientific significance of this study for the management and regulation of hydrological and water resources in Guizhou Province. We have rewritten this sentence. The rewritten sentence is: “This study can provide theoretical support for the optimal regulation and safety management of water resources in karst basins and Guizhou Province under the background of extreme climate change.”. The modified marks can be seen on line 137-139 in the updated PDF version of “Manuscript (revision changes marked)”.